# Combating hypertension beyond genome-wide association studies: Microbiome and artificial intelligence as opportunities for precision medicine

Machine learning; high blood pressure; gut microbiota; genome-based risk scores; personalized medicine

**Corresponding author:**
Bina Joe;
Email: bina.joe@utoledo.edu

S.A. and I.M. contributed equally to this work.

Sachin Aryal[1] , Ishan Manandhar[1], Xue Mei[1], Beng S. Yeoh[1], Ramakumar Tummala[1], Piu Saha[1], Islam Osman[1], Jasenka Zubcevic[1], David J. Durgan[2], Matam Vijay-Kumar[1] and Bina Joe[1]

[1]Center for Hypertension and Precision Medicine, Department of Physiology and Pharmacology, University of Toledo College of Medicine and Life Sciences, Toledo, OH, USA and [2]Integrative Physiology & Anesthesiology, Baylor College of Medicine, Houston, TX, USA

## Abstract

The single largest contributor to human mortality is cardiovascular disease, the top risk factor for which is hypertension (HTN). The last two decades have placed much emphasis on the identification of genetic factors contributing to HTN. As a result, over 1,500 genetic alleles have been associated with human HTN. Mapping studies using genetic models of HTN have yielded hundreds of blood pressure (BP) loci but their individual effects on BP are minor, which limits opportunities to target them in the clinic. The value of collecting genome-wide association data is evident in ongoing research, which is beginning to utilize these data at individual-level genetic disparities combined with artificial intelligence (AI) strategies to develop a polygenic risk score (PRS) for the prediction of HTN. However, PRS alone may or may not be sufficient to account for the incidence and progression of HTN because genetics is responsible for <30% of the risk factors influencing the etiology of HTN pathogenesis. Therefore, integrating data from other nongenetic factors influencing BP regulation will be important to enhance the power of PRS. One such factor is the composition of gut microbiota, which constitute a more recently discovered important contributor to HTN. Studies to-date have clearly demonstrated that the transition from normal BP homeostasis to a state of elevated BP is linked to compositional changes in gut microbiota and its interaction with the host. Here, we first document evidence from studies on gut dysbiosis in animal models and patients with HTN followed by a discussion on the prospects of using microbiota data to develop a metagenomic risk score (MRS) for HTN to be combined with PRS and a clinical risk score (CRS). Finally, we propose that integrating AI to learn from the combined PRS, MRS and CRS may further enhance predictive power for the susceptibility and progression of HTN.

## Impact statement

More than half of the world's adult population suffers from hypertension (HTN), which is the single largest risk factor for human mortality. Despite available medications, susceptibility to develop HTN has not decreased because current knowledge on the risk assessment for susceptibility is severely limited. In this context, genome-wide association studies for HTN are factoring the genetic contributions toward the development of a polygenic risk score (PRS) for HTN. However, given that nongenetic factors also contribute to the etiology of HTN, PRS alone may be insufficient to account for the incidence and progression of HTN. One such nongenetic factor is gut microbiota, which is acquired at birth and demonstrated to be a definitive link to the etiology of HTN. Therefore, here we discuss the prospects for developing and integrating a microbiota-based 'metagenomic risk score' with PRS, and a clinical risk score to construct an artificial intelligence-based model for precision diagnosis and management of HTN.

## Introduction

Despite improvements in health care, cardiovascular disease (CVD) remains the leading cause of human mortality globally (Vos et al., 2020). The propensity to develop CVDs is fueled by chronically elevated blood pressure (BP) or hypertension (HTN). Among others, essential HTN is the most frequent type of HTN in adults (accounts for 95%). It is caused when there is sustained increase in the BP greater than 140/90 mmHg and when no etiology can be

determined for the HTN (Gupta-Malhotra et al., 2015). According to the World Health Organization, an estimated 1.28 billion adults of the age-group 30–79 years worldwide suffer from HTN (https://www.who.int/news-room/fact-sheets/detail/hypertension). Therefore, controlling the incidence of HTN is critical for improving the quality of life and prevention of premature death.

Research on HTN over the last few decades has established that the susceptibility to HTN is determined both by genetic and environmental factors. The estimated contribution of heritability of HTN is ~30%, while environmental factors contribute to ~70% (Biino et al., 2013). Despite the relatively lower contributions of genetics to HTN, there has been considerable focus on mining the genomic contributions to the genesis of HTN. There are two major factors propelling the momentum for understanding the genetics of HTN, (i) the desire to find novel druggable targets and (ii) advances in whole-genome sequencing, which alleviated the technical limitation of detecting human genetic variation on a large scale. Such efforts have thus far identified over 1,500 loci in human HTN (Evangelou et al., 2018; Buniello et al., 2019; Cabrera et al., 2019; Giri et al., 2019). However, while they collectively define the genomic landscape for association with HTN in humans, individually, they are not druggable targets because each of these loci contribute very little to BP regulation.

In experimental studies using animal models, the genomic landscape for association with HTN was similar to that of humans. Animal model studies identified over 400 BP quantitative trait loci (https://rgd.mcw.edu/rgdweb/elasticResults.html?term=blood+pressure&chr=ALL&start=&stop=&species=Rat&category=QTL&objectSearch=true). Details on these investigations are documented in our previous review (Padmanabhan and Joe, 2017) and updated in recent articles (Warren et al., 2017; Evangelou et al., 2018; Giri et al., 2019; Surendran et al., 2020; Olczak et al., 2021; Padmanabhan and Dominiczak, 2021). Meanwhile, research beyond genomic analyses has led to the profound realization that gut microbiota is an important nongenomic factor which was not previously accounted for in the etiology of HTN. Specifically, our group was the first to report the evidence of gut microbiota dysbiosis in both hypertensive animal models and patients (Mell et al., 2015; Yang et al., 2015). Following this pioneering discovery, associations between gut microbiota are reported between hypertensive and normotensive animal models and humans (Tables 1 and 2). In this article, we review the literature on gut microbiota and HTN and propose developing a gut metagenomic risk score (MRS) for HTN. Further, we discuss the value of combining MRS with polygenic risk score (PRS), CRS and artificial intelligence (AI) for clinical management of HTN (Graphical Abstract).

## From GWAS to PRS for HTN

Genome-wide association studies (GWAS) aim to analyze genetic variants across genomes to detect associations with complex traits (Dehghan, 2018). GWAS for HTN began in 2007 with the first report of associations in the Wellcome Trust Case Control Consortium (Burton et al., 2007). GWAS for HTN soon outpaced all linkage analyses in humans (Figure 1b). Even so, the collective effect of all BP loci identified through GWAS accounts for ~3.5% of BP variance (Manolio et al., 2009; Sung et al., 2018). This begs the question: 'What is the expectation from continued investments in GWAS for clinical management of HTN?' From the perspective of disease risk prediction, continued research in GWAS for HTN is essential for developing, defining and refining the predictive power

for HTN using a genomic index, which is known as PRS (Choi et al., 2020; Lewis and Vassos, 2020; Padmanabhan and Dominiczak, 2021). It is computed as the sum of an individual's genome-wide genotype that is weighted by corresponding genotype effect size estimates (or *Z* scores) generated from a relevant GWAS data (Lewis and Vassos, 2020). Although PRSs often explain only a small portion of trait variance, their link with genetic liability, the single biggest source of phenotypic variation, has rendered PRS as an attractive prediction tool in biomedical research (Choi et al., 2020). PRS is used to assess shared etiologies between phenotypes and to investigate the clinical applicability of genetic information for complex diseases (Choi et al., 2020). Previously, a PRS constructed utilizing genome-wide important single nucleotide polymorphisms from GWAS for BP showed a significant relationship with heart failure, left ventricular mass, coronary artery disease and stroke (Studies, 2011; Ference et al., 2014). Currently, there is considerable excitement in the field for developing reliable PRS for HTN as evident from multiple reports of PRS indices from different cohorts (Steinthorsdottir et al., 2020; Sapkota et al., 2021; Sato et al., 2021; Fujii et al., 2022; Parcha et al., 2022; Quintanilha et al., 2022). Recently, Weng et al. (2022) included 391,366 participants from the UK Biobank database and established a PRS for HTN assessing the combined effect of genetic susceptibility and air pollution on incident of HTN. They demonstrated that long-term exposure to air pollution is associated with increased risk of HTN particularly in individuals with high genetic risk (Weng et al., 2022). Another study from Finland revealed that a BP (systolic and diastolic) PRS could predict HTN in the FINRISK cohort, a Finnish population survey on risk factors on chronic, noncommunicable diseases (Vaura et al., 2021, https://thl.fi/en/web/thl-biobank/). This study highlights the potential of PRS as a predictive tool that may be better than the established clinical risk factors for the prediction of HTN (Vaura et al., 2021). But both studies are limited by their reliance on genetic data from European ancestries, which could limit the predictive power of a PRS in other populations. With the availability of recent, large multi-ethnic and non-European GWAS of BP phenotypes, such as those from the Million Veteran Program, the UK Biobank and Biobank Japan (Kanai et al., 2018; Giri et al., 2019), PRS predictions are now expanded to other demographics, which is a promising outlook for the construction of multi-ethnic PRS for HTN risk prediction (Cavazos and Witte, 2021). More recently, the Trans-Omics in Precision Medicine Initiative program (Stilp et al., 2021; Taliun et al., 2021) reported the assessment of PRS for HTN across major U.S. demographic segments. This included African Americans, Hispanic/Latino Americans, Asian Americans and European Americans in the assessment of PRS associations with HTN across the lifespan. The final HTN-PRS was compared with incident outcomes in the Mass General Brigham Biobank as well as with Multi-ethnic Independent Biobank that included 40,201 subjects, leading to associations that supported the links between PRS and HTN. The resulting PRS was also predictive of an elevated risk of type 2 diabetes, chronic renal disease, coronary artery disease and ischemic stroke (Kurniansyah et al., 2022). Based on these results, Kurniansyah et al. (2022) proposed a new approach for tuning parameters for PRS construction including optimization of the coefficient of variation of the effect size estimates and combining PRS based on GWAS of multiple BP phenotypes into a single PRS. Collectively, the next phase of GWAS in HTN should focus on prediction rather than treatment, the implementation of which will depend on the accuracy for applicability of a PRS for HTN in a global setting. To this end, the 'All of Us' research program in the United States is enrolling a million

**Table 1.** The association observed between animal hypertension, gut microbiota and various interventions

*Cambridge Prisms: Precision Medicine*

| References | Model | Intervention | Effect on blood pressure | Enriched/decreased in HTN | Enriched/decreased in normotensive or treatment group |
|---|---|---|---|---|---|
| Yang et al., 2015 | SHR versus WKY | Genetic | BP increased in SHR | ↑ Lactate-producers (*Streptococcus, Turicibacter*) and others *Oribacterium, Parabacteroides* | ↑ Butyrate producers (*Coprococcus, Pseudobutyrivibrio*) and others (*Allobaculum, Bifidobacterium, Alistipes, Blautia* and *Bacteroidetes*) |
| | SD rats | 1. Angiotensin II (Ang II) 2. Ang II + Minocycline | Minocycline attenuated Ang II-induced HTN | *Ang II group:* ↑ F/B ratio, Firmicutes | *Ang II ± Minocycline:* ↑ Acetate and butyrate producers; *Akkermansia, Bacteroides, Enterorhabdus* and *Marvinbryantia* ↓ Bacteroidetes |
| Durgan et al., 2016 | Long Evans Rats | High fat diet (HFD) and Obstructive sleep apnea (OSA) | HFD + OSA increased BP | *HFD ± OSA:* ↑ Coriobacteriaceae, *Lactococcus* ↓ Ruminococcaceae, Clostridaiales | NA |
| | | Cecal content transplant from HTN OSA rats on HFD to normotensive OSA on chow | Microbiotal transplant developed HTN in normotensive OSA on chow | ↑ in Coriobacteriaceae ↓ in Eubacterium | NA |
| Mell et al., 2015 | Dahl salt-sensitive (DSS) and salt-resistant (DSR) rats | High salt diet (HSD) | BP was elevated on HSD | ↑ Bacteroidetes, family S24–7, Lactobacillaceae, Veillonellaceae ↓Enterobacteriaceae | NA |
| | | Cecal content transplant from DSR to DSS versus DSS to DSS transfer | Microbiotal transplant from DSR elevated SBP in DSS rats | ↓ Clostridiales, Veillonellaceae, Mollicutes | NA |
| Santisteban et al., 2017 | Young pre-hypertensive SHR versus young WKY | N/A | Young pre-hypertensive SHR | No significant changes in gut microbiota | NA |
| Santisteban et al., 2017 | Adult SHR versus WKY | N/A | BP elevated in SHR | ↑ Parabacteroides, Porphyromonadaceae, Lactobacillaceae, *Streptococcus*, Streptococcaceae, Latobacillales, *Mogibacterium, Oribacterium, Turibacter* | ↑ Bifidobacterium, Bifidobacteriaceae, Bifidobacteriales, *Gordonibacter*, Bacteroides, Bacteroidaceae, Prevotellaceae, *Alistipes*, Rikenellaceae, Bacteroidales, *Anaerotruncus, Dorea, Blautia, Coprococcus*, Lachnospira, *Allobaculum, Coprabacillus, Massilia*, Oxalobacteraceae, *Escherichia_Shigella*, Enterobacteriaceae, Enterobacteriales |
| Adnan et al., 2017 | WKY rats | FMT from SHRSP | SBP increased by FMT | ↑ Erysipelotrichaceae, Dorea, Anaerostipes, Bacteroidales, Micrococcaceae, *Ruminococcus*, Deferribacterales, Deferribacteres, *Mucispirillum*, Deferribacteraceae, Deferribacteres, *Lactococcus, Desulfovibrio*, Deltaproteobacteria, Desulfovibrionales, *Roseburia, Coprococcus*, Lachnospiraceae, Clostridiales, Firmicutes ↓ Bacteroidetes, Bacteroidia, Erysipelotrichi, Erysipelotrichales, *Allobaculum*, Actinobacteria, Bacteroidaceae, Bacteroides, Actinobacteria, Bifidobacteriales, *Bifidobacterium*, Enterobacteriales, Gammaproteobacteria, Enterobacteriaceae, Betaproteobacteria, | NA |

(Continued)

| References | Model | Intervention | Effect on blood pressure | Enriched/decreased in HTN | Enriched/decreased in normotensive or treatment group |
|---|---|---|---|---|---|
| | | | | *Sutterella*, Alcaligenaceae, Bacillales, Bacillaceae, *Coprobacillus*, Coriobacteriales, Coriobacteriia, *Adlercreutzia*, *Holdemania*, *Enterococcus* | |
| Marques et al., 2017 | C57Bl/6 mice | High-fiber diet and acetate supplementation in Deoxycorticosterone acetate (DOCA) model | SBP and DBP reduced by high-fiber and acetate supplementation in DOCA-induced HTN | *DOCA-control:* ↑ YS2 | ↓ F/B ratio ↑ *Bacteroides acidifaciens*, acetate-producing bacteria |
| Sherman et al., 2018 | Wistar rats | Prenatal androgen (PNA) exposure in induced hypertension in offspring | Increased BP in PNA-exposed rats | *Actinobacteria:* ↑ Yaniellaceae, Geodermatophilaceae, Microbacteriaceae, Nakamurellaceae, Corynebacteriaceae, Promicromonosporaceae and Nocardiaceae ↓ Brevibacteriaceae and Dermabacteraceae *Bacteroidetes:* ↑Rikenellaceae, Paraprevotellaceae ↓ Bacteroidaceae, Odoribacteraceae and *S24–7* were significantly decreased *Firmicutes:* ↑ Peptococcaceae, Eubacteriaceae, Carnobacteriaceae, Tissierellaceae, Streptococcaceae, Veillonellaceae, Coprobacillus and Leuconostocaceae ↓ Ruminococcaceae, Lachnospiraceae, Clostridiaceae, Erysipelotrichaceae, Dehalobacteriaceae, Lactobacillaceae and Mogibacteriaceae *Verrucomicrobia:* ↑ Verrucomicrobiaceae | NA |
| Wilck et al., 2017 | FVB/N mice | High salt diet (HSD) | HSD induced HTN | ↑ *Parasutterella* spp., *Akkermansia* and *Alistipes* ↓ *Lactobacillus, Oscillibacter, Pseudoflavonifractor, Clostridium XIVa, Johnsonella and Rothia* | ↑ Christensenellaceae, Corynebacteriaceae, *Erwinia, Corynebacterium* |
| Robles-Vera et al., 2018 | Wistar rats | NG -Nitro-L-Arginine Methyl Ester (L-NAME) | L-NAME treatment caused progressive increase in SBP and DBP | ↑ F/B ratio, *Propionibacterium* ↓ Parabacteroides, *Bifidobacterium, Olivibacter, Dysgonomonas, Pedobacter, Flavobacterium* and *Desulfotomaculum* | NA |
| Toral et al., 2018 | C57Bl/6 J Mice | Treatment with probiotic *Lactobacillus fermentum* CECT5716 (LC40) in tacrolimus-induced hypertension | Tacrolimus increased SBP and alleviated by LC40 administration | *Tacrolimus group:* ↑ F/B ratio ↓ Butyrate- (including *Butyricimonas*) and acetate-producing bacteria in tacrolimus group, *Bifidobacterium* | ↑ *Bifidobacterium*, (*L. fermentum* CECT5716) ↓ *Anaerostipes*, *Hespellia*, *Prevotella* |
| Bier et al., 2018 | DSS | HSD | BP levels increased in HSD group | ↑ Christensenellaceae, Corynebacteriaceae, *Erwinia, Corynebacterium* | ↑ *Anaerostipes* |
| Waghulde et al., 2018 | DSS | RNA CRISPR (Gper1 deletion) Gper1$^{-/-}$ | *Gper1*$^{-/-}$ rats showed significantly lower BP effect | ↑ unclassified Clostridiales | ↑ Parabacteroides, Bacteroidales S24–7, unclassified enterobacteriaceae ↓ F/B ratio |

*Cambridge Prisms: Precision Medicine*

| References | Model | Intervention | Effect on blood pressure | Enriched/decreased in HTN | Enriched/decreased in normotensive or treatment group |
|---|---|---|---|---|---|
| Galla et al., 2018 | DSS | Neomycin | Neomycin elevated BP | *DSS (neomycin):*<br>↑ Bacteroidetes, Cyanobacteria, Fusobacteria and Verrucomicrobia<br>↓ in Actinobacteria, Deferribacteres, Firmicutes, Proteobacteria, TM7 and Tenericutes | NA |
| | DSS | Minocycline | Minocycline elevated BP | *DSS (minocycline):*<br>↑Firmicutes, Proteobacteria and Verrucomicrobia<br>↓ Actinobacteria, Bacteroidetes, Cyanobacteria, Deferribacteres, TM7 and Tenericutes | NA |
| | DSS | Vancomycine | Vancomycine elevated BP | *DSS (vancomycine):*<br>↑ Bacteroidetes, Cyanobacteria, Elusimicrobia, Fusobacteria, Proteobacteria and Verrucomicrobia<br>↓ Actinobacteria, Deferribacteres, Firmicutes, TM7 and Tenericutes | NA |
| | SHR | Neomycin | No change | *SHR (neomycine):*<br>↑ Bacteroidetes, Cyanobacteria, Elusimicrobia and Verrucomicrobia<br>↓Firmicutes, Proteobacteria, TM7 and Tenericutes | NA |
| | SHR | Minocycline | Reduced BP | *SHR (minocycline):*<br>↑Actinobacteria, Cyanobacteria, Deferribacteres and Firmicutes<br>↓ Bacteroidetes, Proteobacteria, TM7 and Tenericutes | NA |
| | SHR | Vancomycine | Reduced BP | *SHR (vancomycine):*<br>↑Bacteroidetes, Cyanobacteria, Proteobacteria and Verrucomicrobia<br>↓ Firmicutes, TM7, and Tenericutes | |
| Tain et al., 2018 | SD rats | Resveratrol treatment in maternal and post-weaning high fat-induced (HF/HF) HTN | Resveratrol attenuated HF/HF-induced HTN | *HF/HF versus control:*<br>↑ F/B ratio, Verrucomicrobia, *Tepidibacter, Lactococcus, Serratia, Enterobacter, Erwinia, Mucispirillum, Akkermansia municiniphila*<br>↓ Bacteroidetes, *Turicibacter, Lactobacillus, Leuconostoc* | *HF/HF ± Resveratrol:*<br>↑ *Flavobacterium, Tepidibacter, Lactococcus* and *Erysipelothrixas*<br>↓ *Acholeplasma* and *Turicibacter* |
| Sharma et al., 2019 | SD rats | Tetracycline (CMT-3) treatment in Angiotensin II HTN | CMT-3 attenuated Ang II-induced HTN | *Ang II versus control:*<br>↑ Proteobacteria, Parabacteroides, *Blautia*<br>↓ *Ruminococcus* | CMT-3 treatment restored altered taxa in Ang II-induced HTN group |
| Toral et al., 2019b | SHR and WKY | FTM<br>From WKY to SHR | Reduced basal SBP in SHR after FTM from WKY | ↑ Firmicutes<br>↓ Bacteroidetes, F/B ratio | NA |
| Hsu et al., 2019 | SD rats | Maternal and post-weaning high-fat diet (HF/HF) | HF/HF diet elevated BP | *HF/HF:*<br>↑ F/B ratio, Verrucomicrobia, *Akkermansia, Clostridium, Alkaliphilus*<br>↓ *Lactobacillus*, Parabacteroides, *Ruminococcus* | NA |

(*Continued*)

**Table 1.** (*Continued*)

| References | Model | Intervention | Effect on blood pressure | Enriched/decreased in HTN | Enriched/decreased in normotensive or treatment group |
|---|---|---|---|---|---|
| Yang et al., 2019a | SHR and WKY | Captopril (CAP) | CAP decreased BP in SHR | NA | *SHR ± CAP:* ↑ Firmicutes, Proteobacteria, Actinobacteria and Tenericutes, *Mucispirillum*, Parabacteroides, *Allobaculum* |
| Toral et al., 2019a | SHR and WKY | FTM from SHR to WKY (S-W) | Increased basal SBP and DBP | ↑ Odoribacteraceae, *Odoribacter* | NA |
|  |  | FTM from WKY to SHR (W-S) | BP was reduced | NA | ↑ *Blautia*, Peptococcaceae, Lactobacillaceae, *Lactobacillus*, Firmicutes |
| Yan et al., 2020 | Wister rats | HSD | SBP and DBP are significantly higher in HSD group | ↑ F/B ratio, Spirochates ↓ Verrucomicrobia, *Bacteroides fragilis* | N/A |
| Chen et al., 2020 | Sprague–Dawley rats | 20% fructose and 4% Nacl (HFS) | Chronic HFS elevated BP | ↑ Rikenellaceae, Bacteroidetes ↓ F/B ratio, Desulfovibrionaceae | NA |
| Xia et al., 2021 | SHR | Exercise | Exercise significantly decreased SBP in SHR resembling antihypertensive effects | *SHR-exercise versus SHR-sedentary:* ↑ Acetate and Butyrate producers ↓F/B ratio | NA |
| Robles-Vera et al., 2020a | Wister rats | Effect of 1. Bifidobacterium (BFM) 2. Butyrate 3. Acetate in DOCA-salt-induced HTN | DOCA-salt group with BFM treatment prevented the rise in SBP | *DOCA-salt versus control:* ↑ in Actinobacteria, *Blautia*, Peptostreptococcaceae, acetate, butyrate and lactate producers ↓ in Rikinellaceae | *DOCA-salt-BFM group:* ↓ Peptostreptococcaceae resembling the level to that of control group *DOCA-salt-Acetate:* ↑ Bacteroides ↓ *Blautia, Prevotella* |
| Li et al., 2020 | SHR | Maternal captopril effect in SHR offspring | Maternal treatment with captopril significantly lowered BP in SHR Male offspring compared to SHR control. | *SHR:* ↑F/B ratio | *In offspring with maternal route ± sustained CAP:* ↑ Allobaculum, Erysipelotrichaceae and Erysipelotrichia *In offspring with maternal route only:* ↑ Clostridiales, *Anaerostipes, Coprococcus, Oscillospira, Roseburia, Dehalobacterium. SHR offspring with only maternal route of CAP:* ↑ butyrate-producing bacteria, *Coprococcus, Roseburia* and *Oscillospira* |
| Robles-Vera et al., 2020 | WKY | Effect of mycophenolate (MMF) in DOCA-salt-induced HTN | The immunosuppressive drug mofetil MMF significantly reduced BP | *DOCA:* ↑ *Bacilli,* Lactobacillaceae, Burkhholderiales, Betaproteobacteria, Alcaligenaceae | *DOCA ± MMF* ↓ F/B ratio |
| Hsu et al., 2020 | SHR | Effect of maternal *N*-acetylcysteine (NAC) treatment in offspring | Maternal NAC treatment inhibited the rise in SBP | *SHR controls:* ↑ *Bifidobacterium, Lactobacillus, Turicibacter, Akkermansia* ↓ *Holdemania* compared to WKY | *SHR ± NAC:* ↑ Actinobacteria, *Bifidobacterium* and *Allobaculum* ↓ Verrucomicrobia, *Turicibacter,* and *Akkermansia* |
| Robles-Vera et al., 2020b | SHR and WKY | SHR treated with losartan (Angiotensin receptor antagonist) | Losartan (Los) treated SHR showed progressive reduction in SBP | *SHR group versus WKY* ↑ *Lactobacillaceae, Lactobacillus,* and other lactate-producing bacteria ↓ *Verrucomicrobiaceae,* | *SHR ± Los versus SHR* ↓F/B ratio, Verrucomicrobia, ↑ Bacteroidetes, Lactobacillaceae and *Lactobacillus* |

(*Continued*)

| References | Model | Intervention | Effect on blood pressure | Enriched/decreased in HTN | Enriched/decreased in normotensive or treatment group |
|---|---|---|---|---|---|
| | | | | *Pedobacter*, *Akkermansia* and other acetate and propionate-producing bacteria | |
| Galla et al., 2020 | DSS | Amoxicillin administered to young rats | Amoxicillin administration reduced BP | NA | *Amoxicillin versus controls*: ↑ Firmicutes, TM7, Tenericutes, Bacteroidia, Beta Proteobacteria; order: Bacteroidales, Enterobacteriales, and Burkholderiales, Prevotellaceae, Enterococcaceae, Enterobacteriaceae, Alcaligenaceae, Bacteroidaceae, *Blautia*, *Prevotella*, *Enterobacter*, *Enterococcus*, *Klebsiella*, *Sutterella* and *Bacteroides* ↓ in Bacteroidetes, TM7–3, Clostridia, Delta Proteobacteria, Bacilli, Erysipelotrichia, Gamma Proteobacteria and Mollicutes; order: Cw040, Clostridiales, Desulfovibrionales, Lactobacillales, Erysipelotrichales, Pseudomonadales, Turicibacterales and Bacteroidales, F16, Rumincoccaceae, Clostridiales-F, Desulphovibrionaceae, Veillonellaceae, Lacnosphiraceae, Mogibacteriaceae, Lactobacillaceae, Erysipelotrichaceae, Pseudomonadaceae, Clostridiaceae, Turicibacteraceae and Peptostreptococcaceae, *Ruminococcus*, *Oscillospira*, *Anaerovibrio*, (*Ruminococcus*), *Lactobacillus*, *Pseudomonas*, *Coprococcus*, *Dorea*, *Clostridium*, *Roseburia*, *Turicibacter* and (*Prevotella*) (from Paraprevotellaceae family). |
| | | Amoxicillin was administered dams (Gestation and lactation) | Amoxicillin administration reduced BP | | Significantly ↓ in the amoxicillin-treated group These include; classes Delta Proteobacteria and TM7_3; orders Cw040 and Bacteriodales; families Veillonellaceae, F16, Desulfovibrionellaceae and Porphyromonadaceae and genus *Parabacteroide* When compared with their maternal microbiota, amoxicillin consistently ↓ a few groups of bacteria in offspring. F16, Cw040, TM7_3, Veillonellaceae, and Bacteriodales remained consistently reduced in offspring |
| Chakraborty et al., 2020b | DSS | High salt and day/night effect | BP was high during active (night) phase in both high and low salt groups. High SBP in high salt compared to low salt | *High salt (Dark vs. light)*: ↑Sutterella, ↓ Clostridiales *Low salt (Dark vs. light)*: ↓ Streptococcaceae and *Lactobacillus* *High salt dark versus low salt dark*: ↑ *Sutterella,* Erysipelotrichaceae, Ruminococcaceae, ↓Clostridiales *High salt light versus low salt light*: ↓ *Lactobacillus* ↑ Ruminococcaceae | NA |

**Table 1.** (*Continued*)

| References | Model | Intervention | Effect on blood pressure | Enriched/decreased in HTN | Enriched/decreased in normotensive or treatment group |
|---|---|---|---|---|---|
| Abboud et al., 2021 | SHR | Cross-fostered SHR by WKY dams | BP was reduced | *Cross-fostered SHR*<br>↑ *Escherichia-Shigella*<br>↓ *Haemophilus, Lactobacillus intestinalis, Romboustia, Rothia* | NA |
| Shi et al., 2021b | SHRSP and WKY | Every other day fasting (EODF) | EODF attenuated BP rise in SHRSP | *SHRSP Control versus SHRSP EODF:*<br>*SHRSP Control*: ↑ *Bacteroides uniformis, Lactobacillus johnsonii, Lactobacillus reuteri,* Lachnospiraceae bacterium A4, Oscilibacter sp.<br>*SHRSP EODF:*<br>↑ *Asaccharobacter celatus,* Proteobacteria bacterium CAG 139, Muribaculum intestinale, *Parasutterella Excrementihominis* | *WKY Control versus SHRSP Control:*<br>*WKY Control:*<br>↑ *Mucispirillum schaedleri, Oscilibacter_sp_1_3, Bacteroides vulgatus, B. uniformis, Escherichia coli, Parabacateroides goldsteinii, Akkermania Municiniphila*<br>*SHRSP Control:*<br>↑ *Proteobacteria bacterium CAG 139, L. johnsonii, Lactobacillus murinus, A. celatus, Adlercreutzia equolifaciens, Bifidobacterium animalis, Bifidobacterium pseudolongum, Turicimonas muris, Muribaculum intestinale, Parasutterella Excrementihominis* |
| Yang et al., 2022 | SHR | Antibiotics + quinapril | Enhanced BP lowering effect of antibiotics + quinapril | *SHR ± quinapril versus SHR ± quinapril ± Antibiotics:*<br>↑ Lachnospiraceae, *Ruminococcus, Prevotella, Oscillibacter,* Ruminococcaceae UCG_014, Lachnospiraceae UCG_006, *Coprococcus 3, Coprococcus 2. Faecalibaculum, Desulfovibrio, Oscillospira* | NA |
| Shi et al., 2022 | SHRSP | Genetic | SHRSP had elevated SBP at age 8 weeks | *SHRSP versus WKY*<br>↓ Firmicutes, Deferribacterota, *Oscillibacter*<br>↑ Bacteroidota, Verrucomicrobiota, Proteobacteria, *Akkermansia, Allobaculum, Parasutterella* | NA |
| Wu et al., 2022 | SD rats | Effect of captopril in DOCA-induced HTN | SBP was significantly decreased by CAP treatment in DOCA group | *DOCA:* ↑ *Escherichia_Shigella, Eubacterium nodatum group, Ruminococcus_2* | *SHAM:* ↑ *Staphylococcus, Helicobacter, Candidatus Saccharimonas* and *Mucispirillum* and genera *Ruminococcaceae UCG007* and *Peptococcus*<br>*DOCA ± CAP:* ↑ *Bifidobacterium, Victivallis, Akkermansia, Aerococcus, Blautia, Tyzzerella_3* and *Hydrogenoanaerobacterium*<br>↓ Proteobacteria, Cyanobacteria |
| Wang et al., 2022 | SD rats | Cold (4 degree)-induced HTN | Cold expose for 6 weeks induced HTN | *Cold-exposed group:*<br>↑ Bacteroidetes, *Prevotella 1, Quinella, Butyricimonas, Peptococcus, Rothia, Senegalimassilia*<br>↓ Firmicutes | *Control group:*<br>↑ *Lactobacillus,* Lachnospiraceae UCG-008, Ruminococcaceae UCG-013, *Pasteurella,* Lachnospiraceae XPB1014 group, Ruminococcaceae UCG-010, *Coprococcus 3, Lachnospira, Papillibacte, Anaerovorax,* Lachnospiraceae NK4B4 group and *Acetitomaculum* |
| Zheng et al., 2022 | Wistar rats | HSD | HSD-induced HTN | *High salt:*<br>↑ Bacteriodetes, Tenericutes, *B. animalis*<br>↓ Firmicutes<br>*Unique to HSD group* | *Low salt:*<br>↑ *Ruminococcus_2, Butryricoccus, Lactobacillus acidophilus*<br>↓ *Allobaculum*<br>*Unique to low salt group* |

*(Continued)*

**Table 1.** (Continued)

| References | Model | Intervention | Effect on blood pressure | Enriched/decreased in HTN | Enriched/decreased in normotensive or treatment group |
|---|---|---|---|---|---|
| | | | | *Armatimonadetes, Deinococcus-Thermus, FBP, Planctomycetes, Pasteurella multocida, Dubosiella* | *Chloroflexi, Cyanobacteria, Eubacterium plexicaudatam* |
| Hsu et al., 2022 | SD rats | Maternal Tryptophan free diet (TF) | Offspring exposed to maternal TF diet had elevated BP | TF diet group: ↓ Verrucomicrobia, Romboustia, Akkermansia, Ruminococcaceae_NK4A214, Roseburia | NA |
| Mei et al., 2022 | DSS and congenic strain | HSD | Oral Actinobacteria and skin Cyanobacteria are associated with lowering BP | Congenic strains: Skin: ↓ Cyanobacteria in RNO5 female Oral: ↑ Actinobacteria in RNO10 in both sexes | NA |
| Avery et al., 2022 | C57BL/6 J mice | Germ-Free until 4 weeks and then colonization and Angiotensin (Ang) II treatment | Ang II-induced HTN | ↑ Bacteroidetes ↓ Actinobacteria, Akkermansia muciniphila, | NA |

individuals from diverse populations for building a repository that includes genomic data, along with variables such as lifestyle, socio-economic factors, environment and biological factors (All of Us Research Program Investigators, 2019). The United States is a melting pot of diverse populations from around the world. It is therefore particularly interesting to explore this database for further enhancing the power of PRS for HTN.

## Limitations for PRS-based predictions for HTN

Despite the promise and potential of PRS for HTN, there are clear barriers for its application in a clinical setting. One of the main concerns is the environmental component, which has larger effects than the genetic component on BP and may skew the prediction scores. Additionally, PRS analyses are not well-standardized and may lead to faulty interpretations (Choi et al., 2020). Thus, the focus must move from association with case–control status to individualized PRS for enhancing disease prediction (Lewis and Vassos, 2020). Additionally, absolute risks for the disease should be converted from relative risks that compare people across the PRS continuum with a control group (Torkamani et al., 2018; Sugrue and Desikan, 2019). When using PRS for HTN prediction, management and treatment, it is also required to rigorously differentiate between essential HTN and secondary HTN. Finally, as is the case with all diseases, there are ethical concerns regarding the application of PRS for HTN, which may escalate health inequities (Minari et al., 2018; Martin et al., 2019; Vaura et al., 2021).

## Progress beyond GWAS: What are we missing?

As mentioned above, the premise of using PRS alone for HTN lacks power because of the environmental factors contributing to its etiology. In this context, it is important to note that a prominent, previously unknown, and relatively recent factor identified as contributing to BP regulation is the composition of gut microbiota. As shown in Figure 1a,b, the numbers of studies on microbiota and HTN is sharply rising in both animal models and humans. Interestingly, the sheer numbers of such studies currently surpasses that of GWAS studies, indicating its importance. In the following sections, we review these studies and propose that the inclusion of microbiota signatures and their functional readouts along with the genetic makeup of the host may enhance the power of PRS for HTN.

## Gut microbiota and HTN

A large body of evidence has emerged in the last decade supporting the role of the microbiota in BP regulation. Our group has been at the forefront of this research. In 2010, it was shown that knockout of toll-like receptor 5 (Tlr5) in mice resulted in elevated BP (Vijay-Kumar et al., 2010). Tlr5 is a receptor for the bacterial protein flagellin, suggesting a link between gut microbiota and HTN. However, the major focus of this report was on metabolic syndrome, of which BP is a hallmark. The first evidence for a direct link between gut microbiota and BP regulation in a genetic model of HTN was reported in 2015 in Dahl salt-sensitive (DSS) rat (Mell et al., 2015). Shortly thereafter, an association between gut dysbiosis and HTN in spontaneously hypertensive rats (SHR), angiotensin II-induced hypertensive rats, sleep apnea-induced hypertensive rats (Lloyd et al., 2015) and hypertensive humans

**Table 2.** The association observed between human hypertension, gut microbiota and various interventions

| References | Model | Intervention | Taxa positively associated with BP | Taxa negatively associated with BP |
|---|---|---|---|---|
| Munukka et al., 2012 | 74 premenopausal women and 11 healthy females | NA | *Genera*:<br>*Proportion of Eubacterium rectale- Clostridium coccoides* (compared to nonmetabolic disorder group and normal-weight women group) | NA |
| Queipo-Ortuño et al., 2012 | 10 healthy male volunteers | Red wine polyphenols and ethanol | *Genera*:<br>*Enterococcus, Prevotella, Bacteroides, Bifidobacterium, Bacteroides uniformis, Eggerthella lenta, Blautia coccoides-E. rectale groups* | NA |
| Gomez-Arango et al., 2016 | 86 overweight versus 119 obese women | 16 weeks gestation | NA | *Genera*:<br>*Blautia, Odoribacter*, families Odoribacteraceae Clostridiaceae Christensenellaceae. (More blood pressure in obese women) |
| Li et al., 2017 | 41 healthy controls, 56 pre-hypertensives, 99 primary hypertensives | Fecal microbiota transplant | *Genera*:<br>*Prevotella, Klebsiella, Porphyromonas, Actinomyces, Desulfovibrio, Fusobacterium* | *Genera*:<br>*Bacteroides, Faecalibacterium, Oscillibacter, Roseburia, Bifidobacterium, Coprococcus, Butyrivibrio, Clostridium, Enterococcus, Blautia* |
| Yan et al., 2017 | 60 primary hypertensives and 60 controls | NA | *Genera*:<br>*Klebsiella, Clostridium, Streptococcus, Parabacteroides, Eggerthella, Salmonella*<br>*Phylum*: Proteobacteria | *Genera*: *Faecalibacterium prausnitzii, Roseburia* and *Synergistetes*<br>*Phylum*: Actinobacteria |
| Wilck et al., 2017 | 12 healthy males | Dietary salt | NA | *Genera*:<br>*Lactobacillus* spp. |
| de la Cuesta-Zuluaga et al., 2018 | 441 men and women | Fecal SCFA excretion | *Genera*:<br>*Enterobacter hormaechei, Haemophilus parainfluenzae, Streptococcus, SMB53* | NA |
| Kim et al., 2018 | 22 hypertensives and 18 reference cohorts | NA | *Genera*:<br>*Alistipes finegoldii, Dorea, Alistipes indistinctus* | *Genera*:<br>*E. rectale, Bacteroides thetaiotaomicron, Klebsiella, Burkholderiales bacterium, Burkholderiales noname, Paraprevotella xylaniphila, Bacteroides salyerside, Veillonella, Paraprevotella clara, Ruminococccus callidus* |
| Liu et al., 2018 | 94 hypertensives and 94 healthy controls | NA | *Genera*:<br>*E. rectale*<br>*Phylum*:<br>Firmicutes | *Genera*:<br>*B. thetaiotaomicron, Bifidobacterium* |
| Jackson et al., 2018 | 756 hypertensives and 1,790 controls | Prescription medication | *Family*:<br>Lactobacillaceae, Streptococcaceae, Enterococcaceae | *Family*:<br>Dehalobacteriaceae, Christensenellaceae, Oxalobacteraceae, Rikenellaceae, Clostridiaceae, Anaeroplasmataceae, Peptococcaceae |
| Ried et al., 2018 | 49 participants with uncontrolled hypertension | Kyolic aged garlic extract supplement | NA | *Genera* (increased in garlic supplement group compared to placebo):<br>*Lactobacillus*<br>*Clostridia* species<br>*Genera* (increased in placebo compared to garlic supplement group): *Faecalibacterium prausnitzi* |

(*Continued*)

*Cambridge Prisms: Precision Medicine*

| References | Model | Intervention | Taxa positively associated with BP | Taxa negatively associated with BP |
|---|---|---|---|---|
| Han et al., 2018 | 99 nontreated hypertensives, 56 pre-hypertensives and 41 normotensives | NA | *Genera*:<br>*Prevotella copri* in HTN and pre-HTN group<br>Cronobacter phage CR3 in pre-HTN<br>Cnaphalocrocis medinalis granulovirus in HTN group | Streptococcus virus phiAbc2, Salmonella phage vB SemP Emek, Mycobacterium phage Toto |
| Dan et al., 2019 | 67 normotensives<br>62 hypertensives | NA | *Genera*:<br>*Acetobacteroides, Alistipes, Bacteroides, Barnesiella, Christensenella, Clostridium sensu stricto, Cosenzaea, Desulfovibrio, Dialister, Eisenbergiella, Faecalitalea, Megasphaera, Microvirgula, Mitsuokella, Parabacteroides, Proteiniborus, Terrisporobacter* | *Genera*:<br>*Anaerotruncus, Prevotella, Oscillibacter, Butyricimonas, Acetobacteroides, Acidaminobacter, Adlercreutzia, Anaerotruncus, Asteroleplasma, Bulleidia, Cellulosilyticum, Clostridium III, Clostridium IV, ClostridiumXlVa, Coprobacter, Enterococcus, Enterorhabdus, Flavonifractor, Gemmiger, Guggenheimella, Intestinimonas, Lachnospiracea_incertae_sedis, Lactivibrio, Lactobacillus, Macellibacteroides, Marvinbryantia, Olsenella, Paraprevotella, Parasutterella, Phascolarctobacterium, Prevotella, Romboutsia, Ruminococcus, Sporobacter, Sporobacterium, Sutterella, Vampirovibrio, Veillonella, Victivallis* |
| Sun et al., 2019 | 183 hypertensives and 346 normotensives. | NA | *Genera*:<br>*Robinsoniella, Catabacter*<br>*Family*:<br>Veillonellacaeae | *Genera*:<br>*Sporobacter, Anaerovorax, Ruminococcus*<br>*Family*:<br>Ruminococcaceae |
| Li et al., 2019b | 63 hypertensives with treatment-naïve hypertension, 104 hypertensive patients undergoing antihypertensive treatment, 26 normal bp patients with hyperlipidemia, 42 healthy controls | NA | *Genera*:<br>*Megamonas, Megasphaera, Lactococcus, Alistipes, Subdoligranulum* | *Genera*:<br>*Clostridium sensu stricto 1, Romboutsia, Erysipelotrichaceae UCG.003, Ruminococcus 2, Intestinibacter* |
| Mushtaq et al., 2019 | 50 patients with grade 3 hypertension, 30 healthy controls | NA | *Genera*:<br>*Prevotella_9, Megasphaera, Parasutterella, Escherichia-Shigella, Phascolarctobacterium faecium* | *Genera*:<br>*F. prausnitzii, B. uniformis* |
| Huart et al., 2019 | 38 hypertensives, 7 borderline and 9 normotensives | 21 hypertensives under antihypertensive medication | *Genera*:<br>*Clostridium sensu stricto 1* | *Genera*:<br>*Ruminococcaceae_ge_ DQ807686, Clostridiales_ge_16S_ OTU1343* |
| Bellikci-Koyu et al., 2019 | 12 Kefir consuming group and 10 unfermented milk consuming group. All hypertensives | Kefir and unfermented milk consumption | *Phylum*:<br>Bacteroidetes | *Genera*:<br>*Lactobacillus*<br>*Bifidobacterium* spp<br>*Phylum*:<br>Actinobacteria<br>Firmicutes<br>Verrucomicrobia |
| Ferguson et al. n.d. | 39 subjects with normal salt intake and 93 with high sodium intake | Dietary salt | *Genera*:<br>*Prevotella, Bacteroides*<br>*Family*:<br>Ruminococcaceae<br>*Phylum*:<br>Firmicutes, Proteobacteria | NA |

**Table 2.** (*Continued*)

| References | Model | Intervention | Taxa positively associated with BP | Taxa negatively associated with BP |
|---|---|---|---|---|
| Verhaar et al., 2020 | 1,937 hypertensives and 2,735 normotensives | Fecal SCFAs | *Genera:*<br>*Klebsiella* spp., *Streptococcus* | *Genera:*<br>*Roseburia* spp., *Reseburia hominis, Ruminococcaceae, Clostridium sensu stricto 1, Romboutsia, Enterorhabdus* |
| Shah et al., 2020 | ABO study (Enterotype 1; *n* = 53 and Enterotype 2; *n* = 78) and Fair study (*N* = 29) | Dietary soy | *Genus*:<br>*Prevotella, Dialister*<br>(in Enterotype 1) | NA |
| Palmu et al., 2020 | 3,291 hypertensives and 3,662 normotensives | Dietary sodium | *Genera*:<br>*Anaerostipes, Anaerotruncus, Bacteroides, Blautia, Citrobacter, Colinsella, Coprobacillus, Coprococcus, Dielma, Dorea, Eisenbergiella, Enterobacter, Erysipelatoclostridium, Faecalitalea, Fournierella, Holdemania, Intestinibacter, Kluyvera, Lactococcus, Megasphaera, Phascolarctobacterium, Ruthenibacterium, Mitsuokella, Paraprevotella, Sanguibacteroides, Sutterella, Turicibacter, Acidaminococcus, Actinomyces, Lactobacillus salivarius* | *Genera*:<br>*Lactobacillus paracasei, Adlercreutzia, Alloprevotella, Anaerotruncus, Coprobacillus, Faecalicoccus, Fournierella, Hungatella, Parasutterella, Prevotella, Sellimonas, Senegalimassilia, Solobacterium, Tyzzerella* |
| Chang et al., 2020 | 27 Preeclampsia (PE) and 36 healthy pregnant control | NA | *Genera (in PE):*<br>*Enterobacter, Escherichia_Shigella*<br>*Phylum*:<br>Proteobacteria | *Genera (in PE):*<br>*Blautia, Eubacterium_rectale, Eubacterium_halii, Streptococcus, Bifidobacterium, Collinsella, Alistipes, Subdoligranulum*<br>*Phylum*:<br>Firmicutes |
| Tindall et al., 2020 | 42 cardiovascular risk adults | Standard western diet run-in and isocaloric study diets | *Genera*:<br>After walnut diet (relative to standard western diet) characterized by<br>an elevated blood pressure, LDL cholesterol and BMI<br>*Roseburia, Eubacterium eligensgroup, Lachnospiraceae UCG001, Lachnospiraceae UCG004, Leuconostocaceae*<br>Relative to walnut fatty acid-matched diet<br>*Gordonibacter*<br>After the walnut fatty acid-matched diet<br>(relative to standard western diet)<br>*Roseburia, E. eligensgroup*<br>After the oleic acid replaces α-linolenic acid diet<br>(relative to standard western diet)<br>*Clostridialesvadin BB60group* | *Genera*:<br>After the whole walnut diet<br>*Lachnospiraceae* (inversely correlated with blood pressure) |
| Silveira-Nunes et al., 2020 | 48 hypertensives and 32 normotensives | NA | *Genera*:<br>*L. salivarius, Eggerthella, Bacteroides plebeius* | *Genera*:<br>*Roseburia faecis, F. prausnitzii, Fusobacterium, Coprococcus* |
| Takagi et al., 2020 | 54 controls, 97 hypertensives 96 hyperlipidemia and 162 type 2 diabetes mellitus | NA | *Genera*:<br>*Collinsella*<br>*Bifidobacterium*<br>*Phylum*:<br>Actinobacteria | *Phylum*:<br>Bacteroidetes |
| Capper et al., 2020 | 36 healthy participants (19 beetroot group and 17 control group) | Whole beetroot consumption in older population | *Genera*:<br>*Prevotella_9*<br>*Phylum*:<br>Bacteroidetes | *Genera*:<br>*Alistipes, Faecalibacterium, Akkermansia Christensenellaceae_R-7 group* |

(*Continued*)

**Table 2.** (*Continued*)

| References | Model | Intervention | Taxa positively associated with BP | Taxa negatively associated with BP |
|---|---|---|---|---|
| Calderón-Pérez et al., 2020 | 29 nontreated hypertensives and 32 normotensives | NA | *Genera*: *Bacteroides coprocola, B. plebeius* and genera of *Lachnospiraceae* | *Genera*: *Ruminococcaceae NK4A214, Ruminococcaceae_UCG-010, Christensenellaceae_R-7, F. prausnitzii, Roseburia hominis* |
| Wan et al., 2020 | First day versus 2 weeks versus 6 weeks postpartum milk from 117 mothers | Mother's milk | NA | *Lactobacillus* is lower in 2 weeks milk in gestational pre-hypertensive mothers. |
| Louca et al., 2021 | 397 hypertensives and 474 normotensives | NA | NA | *Genera*: *Ruminiclostridium 6* |
| Wang et al., 2021 | 93 hypertensives and 15 healthy controls | Effect of electroacupuncture in healthy and HTN group | *Genera*: *Escherichia-Shigella* Firmicutes/Bacteroidetes lowered after treatment | *Genera*: *Blautia* |
| Liu et al., 2021b | 13 primary aldosteronism (PA) patients, 26 sex-matched primary hypertensives, and 26 sex-matched healthy controls | PA and PHTN patients received antihypertensive medications before recruitment. | *Genera*: *Megamonas, Sutterella Lactobacillus, Enterococcus Bacillus, Bifidobacterium, Phascolarctobacterium, Pseudomonas, Weissella, Ruminococcus gnavus group, Pediococcus, Acinetobacter, Lactococcus, Akkermansia, Alloprevotella, Staphylococcus, Wolbachia, Halomonas, Bradyrhizobium* | *Genera*: *Faecalibacterium, Subdoligranulum, Roseburia, Coprococcus, Blautia, Ruminococcus, Agathobacter, Alistipes, Adlercreutzia, Paraprevotella, Erysipelotrichaceae UCG 003 Christensenellaceae R-7 group Eubacterium ventriosum group E. eligens group* Firmicutes/Bacteroidetes ratio *Phylum*: Proteobacteria *Family*: Lachnospiraceae |
| Wan et al., 2021 | 300 healthy controls, 300 hypertensives and 300 coronary heart disease patients | NA | *Genera*: Enterobacteriales *Escherichia shigella* | *Genera*: Acidaminococcaeceae *Phascolarctobacterium Phylum*: Bacteroidetes, Bacteroidia |
| Zhong et al., 2021 | 73 hypertensives and 187 normotensives | Washed microbiota transplant from control to HTN | *Genera*: *Parasutterella. Solobacterium* | *Genera*: *Senegalimassilia* |
| Calderón-Pérez et al., 2021 | 29 hypertensives versus 32 normotensives | Phenolic compound (sources such as coffee, olive fruits and vegetables) | *Genera*: *Bacteroides coprocola Bacteroides plebeius* | *Genera*: *Ruminococcaceae NK4A214, Ruminococcaceae UCG-010, Christensenellaceae R-7, F. prausnitzii* |
| Joishy et al., 2022 | 37 normotensives and 34 hypertensives | NA | *Genera*: *Prevotella* (different ASVs)*, Megasphaera, Butyricoccus, Prevotellaceae, Faecalibacterium, Lachnoclostridium, Howardella* and g-UCG04 | *Genera*: *Prevotella* (other ASVs)*, Alloprevotella, Streptococcus* |

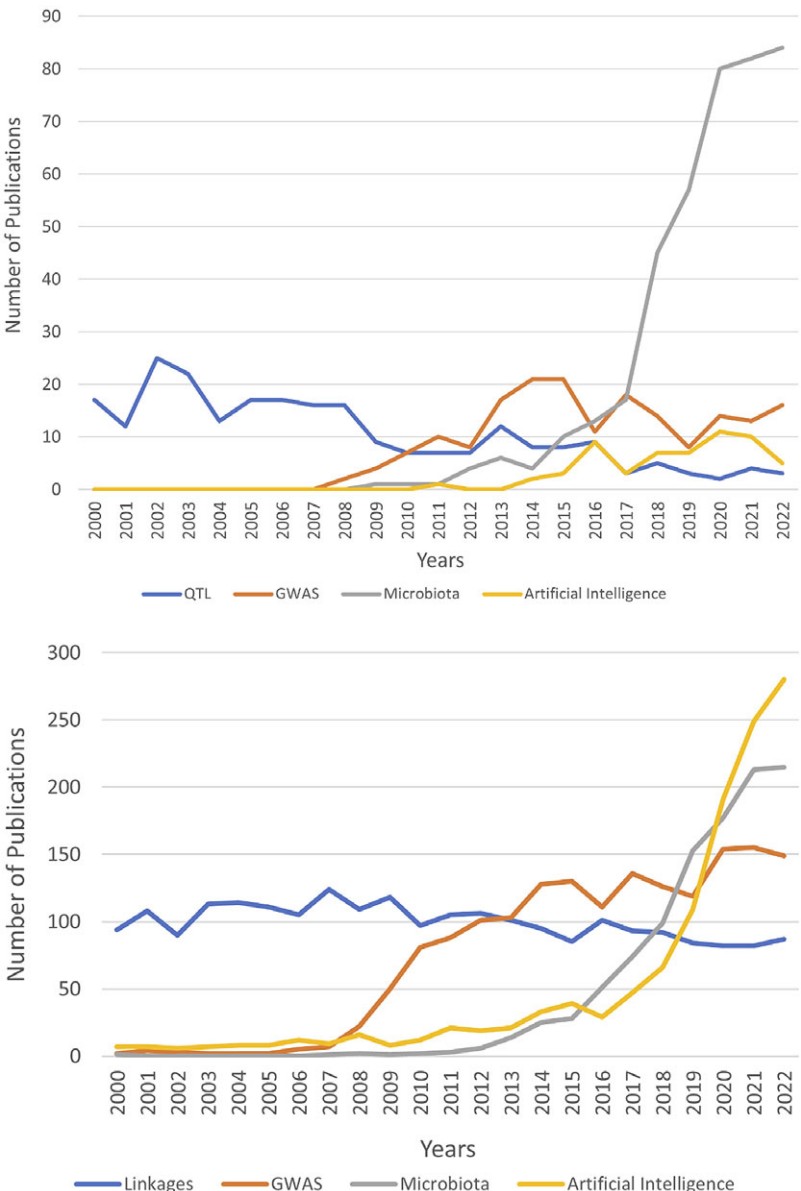

**Figure 1.** (a) The numbers of PubMed publications (2000–2022) related to quantitative trait locus (QTL), genome-wide association studies (GWAS), microbiota, artificial intelligence in rats and mice hypertension. The search keywords were QTL, hypertension, rats, mice, GWAS, microbiota and artificial intelligence. (b) The numbers of PubMed publications (2000–2022) related to linkage, genome-wide association studies (GWAS), microbiota and artificial intelligence in human hypertension. The search keywords were linkage, hypertension, humans, GWAS, microbiota and artificial intelligence.

were reported (Yang et al., 2015). Since these initial groundbreaking reports, multiple publications have demonstrated associations of gut microbiota with BP regulation in animal models and humans (Tables 1 and 2).

One question of assessing microbiota composition in rats is whether they are translationally relevant for humans. Human gut microbiota composition is more similar to rats than to mice (Flemer et al., 2017), although this question is evolving with continued development of sequencing methods. Here, we access the commonalities in taxonomic rearrangements occurring during HTN in rats and humans. One of the themes emanating from BP association studies using rat as model organism is the application of the Firmicutes/Bacteroidetes (F/B) ratio in assessment of gut dysbiosis in HTN. Increased F/B ratio is regarded as a marker of gut dysbiosis and is consistently reported both in genetic and induced

hypertensive rat models including the SHR (Yang et al., 2015; Hsu et al., 2020; Li et al., 2020), DSS rats (Mell et al., 2015; Waghulde et al., 2018) high-fat diet fed rats (Hsu et al., 2019), N^G-nitro-L-arginine methyl ester (L-NAME) treated rats (Robles-Vera et al., 2018) and angiotensin II induced HTN rats (Yang et al., 2015). In further support, normalizing the F/B ratio by administration of the anti-inflammatory antibiotic, minocycline, alleviated angiotensin II-induced HTN (Yang et al., 2015). This direct relationship between F/B ratio and BP has also been documented in various mouse models (Marques et al., 2017; Toral et al., 2018). Similarly, human studies (Mushtaq et al., 2019; Silveira-Nunes et al., 2020; Joishy et al., 2022) also support a direct relationship between F/B ratio and BP. In contrast, cold-induced HTN (Wang et al., 2022) is one of the rare contexts wherein F/B ratio was not altered significantly. Nevertheless, more robust markers of gut dysbiosis should

be developed to posit strong correlation between decreased microbial diversity and HTN.

Beyond the F/B ratio, remodeling of the overall composition of gut microbiota has been documented in the context of HTN. For example, enrichment in gut bacterial lactate producers such as *Streptococcus* and *Turicibacter* (Yang et al., 2015; Toral et al., 2019b; Robles-Vera et al., 2020b) and depletion of butyrate producers such as *Coprococcus,* and *Pseudobutyrivibrio* (Yang et al., 2015; Durgan et al., 2016) are reported in hypertensive rodents. Importantly, *Streptococcus* and *Coprococcus* are also taxa similarly associated with human HTN (Yan et al., 2017; de la Cuesta-Zuluaga et al., 2018; Palmu et al., 2020; Verhaar et al., 2020), and normalizing abundance of these with minocycline and captopril (Yang et al., 2015; Li et al., 2020) lowered BP, further supporting their associations with BP.

Dietary interventions have also been used to study the relationship between gut microbiota and BP. Dietary salt can modulate the composition of microbiota by depleting the abundance of beneficial microbiota including several Lactobacilli species (Mell et al., 2015; Wilck et al., 2017; Bier et al., 2018; Yan et al., 2020). An association between the depletion of *Lactobacillus* and HTN has also been noted in response to maternal and post-weaning high-fat diet, and it suggested that *Lactobacillus* may be beneficial in curbing developmental HTN (Tain et al., 2018; Hsu et al., 2019). Although the mechanisms remain to be clarified, administration of *Lactobacillus murinus* prevented the expansion of proinflammatory IL-17A-producing CD4[+] $T_H$17 lymphocytes in small intestine, colon, and the splenic lamina propria (Wilck et al., 2017). Data from human studies with *Lactobacillus* are however conflicting, as they may be enriched or depleted in hypertensive patients (Wilck et al., 2017; Palmu et al., 2020; Silveira-Nunes et al., 2020; Wan et al., 2020; Liu et al., 2021b).

Gut metabolites, derived either from gut microbiota or involving both gut microbiota and host is of growing interest in the context of HTN. One such important class of microbial metabolites is short-chain fatty acid (SCFA). SCFAs such as acetate, propionate and butyrate are produced by bacterial fermentation of dietary carbohydrates and have been linked with BP regulation. It is reported that decreased SCFA production and the supplementation of SCFA lowered BP in rat and mouse HTN models, indicating the potential for antihypertensive therapy (Marques et al., 2017; Kim et al., 2018; Bartolomaeus et al., 2019; Robles-Vera et al., 2020c). In deoxycorticosterone acetate (DOCA)-salt-induced HTN, high-fiber diet lowered BP and enriched abundance of gut microbes producing acetate (Marques et al., 2017). Interestingly, higher fecal levels of SCFA were associated with hypertensive individuals compared to normotensives (de la Cuesta-Zuluaga et al., 2018; Huart et al., 2019; Calderón-Pérez et al., 2020). Further, increased fecal SCFA was accompanied by decreased plasma SCFA and depleted butyrate-producing bacteria which suggests dysregulated production of SCFA in HTN condition (Calderón-Pérez et al., 2020). The translational relevance and progress of animal as well as clinical studies in SCFA and BP have resulted in a clinical trial to determine the full efficacy of SCFA to treat HTN (Australian New Zealand Clinical Trials Registry ACTRN12619000916145). This phase II clinical trial used two SCFAs, acetate and butyrate which were supplemented with high-amylose maize, and the patients receiving the treatment showed 24-h BP lowering effect with the increase in gut microbes producing SCFA (Jama et al., 2023), which is another evidence of the promising potential of targeting gut microbiota in HTN treatment. Another notable microbial metabolite is trimethylamine-*N* oxide (TMAO). Trimethylamine is produced by gut microbiota, and subsequently oxidized in the liver to form TMAO. Studies have shown the associations between higher plasma levels of TMAO and CVDs (Koeth et al., 2013, 2019; Wang et al., 2014). In a meta-analysis involving human studies, higher circulating TMAO concentration was positively associated with an increased risk of HTN (Ge et al., 2020). TMAO feeding further increased BP and promoted vasoconstriction in angiotensin II-induced hypertensive mice (Jiang et al., 2021). Besides SCFA and TMAO, there are many microbiota-derived metabolites such as indole, indole-3-acetic acids and secondary bile acids among others, that may have significant roles in BP regulation (Huć et al., 2018; Chakraborty et al., 2020a). The knowledge on the effects of these microbial metabolites is growing, however, the precise underlying working mechanisms remain largely unknown.

## Gut microbiota restructured in HTN: Cause, consequence or adaptation?

While an association between the reprogramming of gut microbiota and HTN is established, whether gut dysbiosis is a cause or a consequence of HTN is an important question to focus on. Initial experiments were designed to address this question by using antibiotics to eliminate endogenous gut microbiota. However, such studies did not provide conclusive evidence for cause or consequence because different antibiotics affected BP differently depending on both the type of antibiotic and the rodent strain (Galla et al., 2018, 2020). More convincing evidence for microbiota to cause a BP effect was obtained using germ-free Sprague Dawley (SD) rats. We showed that these rats which lack microbiota are hypotensive, with a prominent loss of vascular tone (Joe et al., 2020). These findings are the first to clearly demonstrate that the host requires gut microbiota for BP homeostasis and maintenance of vascular tone (Joe et al., 2020). One caveat to these studies is that the model used is not hypertensive. To establish the cause-effect relationship between HTN and gut microbiota, animal models such as germ-free hypertensive rat models can be developed. Such hypertensive germ-free rats will allow for testing the hypothesis that lack of microbiota will render them resistant to HTN. Currently, the lack of germ-free hypertensive rats as tools is a technical barrier to understand whether microbiota cause HTN.

To examine the causality of gut dysbiosis, Adnan et al. (2017) performed gut microbiota cross-transplants between WKY and spontaneously hypertensive stroke-prone rats (SHRSP, a rodent model of HTN associated with high incidence of stroke. They observed that a stable transplant of SHRSP gut microbiota to normotensive WKY recipients, by oral gavage, led to a significant elevation in BP (Adnan et al., 2017). Similar trend was observed in another study after fecal microbiotal transplantation from SHR to WKY (Toral et al., 2019b). As an alternative approach to oral gavage transplants, which involve exposure to antibiotics, Nelson et al. (2021) swapped WKY and SHRSP gut microbiota using a cross-fostering protocol. By fostering newborn rat pups with a dam of the opposite strain, SHRSP rats were populated with a WKY gut microbiota and vice versa. Under these conditions, WKY rats harboring SHRSP gut microbiota developed a significantly elevated BP in adulthood compared to WKY rats with native WKY microbiota. Conversely, adult SHRSP rats harboring the WKY gut microbiota presented with significantly lower BP compared to SHRSP with their native SHRSP microbiota (Nelson et al., 2021). These data signify that initial colonization of gut microbiota is critical and has long-lasting consequences on host pathophysiology.

As highlighted above, research focus has shifted to better understanding the mechanisms of host-microbiota interactions. Emerging studies address molecular mechanisms and demonstrate how BP may be regulated by bacterial metabolites via effects on the aryl hydrocarbon receptor (Natividad et al., 2018; Liu et al., 2021a), and G-protein coupled receptors (Marques et al., 2018; Xu and Marques, 2022) among others, which may impact end organ functions of the kidney, vasculature, brain and heart. Emerging work from Durgan et al. shows that a new mechanism by which signals derived from the gut microbiota (i.e., metabolites, neurotransmitters, endotoxins) may be distributed throughout the host via packaging into outer membrane vesicles (OMVs). These OMVs are lipid-bound vesicles (as known as bacterial liposomes) released from the gut microbiota that are capable of crossing the gut barrier and entering the systemic circulation. Bacterial OMVs can carry a wide range of 'cargo' including proteins, lipids, and small RNA, that can be delivered to and exert effects on distant host cells. They have shown that OMVs from the SHRSP microbiota have unique protein and lipid cargo as compared to OMVs from the WKY microbiota. Additionally, they find that SHRSP OMVs gavaged to WKY rats leads to significant elevations in BP (Shi et al., 2021a).

Previous reports showed no gut dysbiosis in pre-hypertensive SHR (Santisteban et al., 2017; Yang et al., 2020), suggesting that gut dysbiosis may arise as a consequence of HTN. However, these studies demonstrated colonic changes in pre-hypertensive SHR indicating a dysregulated gut barrier before developing HTN. Future studies should address this more specifically. Nevertheless, transplant experiments show that gut dysbiosis contributes to HTN and that manipulation of gut microbiota can alleviate HTN, suggesting that gut microbiota could be a potential therapeutic target.

## Gut microbiota as therapeutic targets

There is considerable excitement of targeting gut microbiota for translational applications as evident from ongoing clinical trials for microbiota-guided therapies for HTN (https://clinicaltrials.gov/ct2/results?cond=hypertension&term=minocycline&cntry=&state=&city=&dist=). In preclinical studies, our group recently proposed *Faecalibacterium prausnitzii* as a novel probiotic to attenuate chronic kidney disease (CKD) conditions, following demonstrated depletion of *F. prausnitzii* in CKD patients in eastern and western human hypertensive populations. Importantly, supplementation of *F. prausnitzii* in a CKD mouse model not only ameliorated renal dysfunction, renal inflammation, and the levels of uremic toxins, but also improved gut ecology and intestinal integrity (Li et al., 2022). Since *F. prausnitzii* has also been reported to be depleted in CVD (Jie et al., 2017; Aryal et al., 2020), and CVD and CKD are highly correlated, it is possible that enhancing *F. prausnitzii* could be beneficial for CVD, for which HTN is a major risk factor (Li et al., 2022). Supporting this notion, studies have shown that *F. prausnitzii* is significantly abundant in normotensive compared to hypertensive humans, demonstrating strong correlation of this specific microbe with BP (Yan et al., 2017; Calderón-Pérez et al., 2020). However, in contradiction, *Faecalibacterium* was more enriched in individuals with high BP (Joishy et al., 2022). Therefore, there is a need to directly examine the potential of *F. prausnitzii* in rigorous animal model studies.

In addition to being considered as therapeutic agents, gut microbiota may be involved in the modulation of our responses to antihypertensive medications (Kyoung et al., 2022). The efficacy of angiotensin-converting enzyme (ACE) inhibitors is reportedly modulated by the gut microbiota (Kyoung and Yang, 2022; Yang et al., 2022). Our group has recently demonstrated that quinapril, which is absorbed in the gut and metabolized by esterases in the liver to yield an active metabolite in circulation is prematurely catabolized in the gut by microbiota. This led to reduced availability of the active metabolite, quinaprilatin in circulation, which was associated with reduced BP responses to oral administration of quinalapril (Yang et al., 2022). We further identified that a specific microbiota, *Coprococcus comes*, contains a bacterial form of esterase and may be one of the culprits for the premature quinalapril degradation and reduction in its efficacy as a BP-lowering agent. Interestingly, a higher abundance of *C. comes* is present in the African American hypertensive population (Yang et al., 2022) who are known to respond poorly to ACE inhibitor treatments compared to Caucasian hypertensive patients (Yang et al., 2022). This proof-of-concept study implicates that gut microbiota is a crucial factor defining individualized responses to hypertensive medications that should be addressed in future efficacy studies of antihypertensive drugs.

## Current limitations in microbiome research for HTN

While such physiological studies are clearly important, drawing conclusions about the role of individual microbes in BP regulation, based on 16S analysis alone, can be problematic. The issue stems from the fact that multiple species can carry out the same function (e.g., generate the same metabolite), also referred to as functional redundancy. This redundancy likely contributes to the disparate candidate bacteria identified across hypertensive models and research facilities (Table 1). Another limitation of 16S analysis is that it captures a limited portion of the bacterial genome (Lewis et al., 2021). These limitations can be addressed by sequencing of whole bacterial genomes known as metagenomic sequencing, which is an emerging area in HTN research (Walejko et al., 2018; Shi et al., 2021b). It should be noted that at the current stage, it is a misnomer to use the term 'microbiome' until metagenomes are reported. With the advent of rapid and cost-effective technologies, progress in reporting of metagenomes is anticipated to provide a platform for association studies of metagenomes with HTN. Metagenomics however falls short in assessing activity of the identified bacterial genes. Thus, combining metagenomics with an assessment of the functional output from the microbiota (i.e., proteomics, metabolomics, lipidomics) can be especially powerful. A recent study by the Durgan laboratory examined the role of gut dysbiosis in the SHRSP model by combining metagenomics with metabolomics analysis of the cecal bacterial content and the host plasma. While metabolomics revealed significant reductions in cecal and plasma primary and secondary bile acids in the SHRSP, metagenomics pinpointed that specific genes encoding bacterial enzymes involved in bile acid transformation were also reduced in the SHRSP microbiota. Thus, assessing changes in microbiota function will be useful in the development of targeted approaches for the treatment of HTN.

One other limitation in elucidation of functional consequences of host-microbiota interactions is addressing the complexity of such interactions. Gut microbiota and the host have evolved for centuries to live in complete symbiosis. This means that they are mutually dependent for survival and homeostasis. For example, humans are not capable of degrading fiber (Kaoutari et al., 2013; Cockburn and Koropatkin, 2016). Gut bacteria aid the host by

fermentation of fiber, thus generating SCFAs which are the main source of energy for the host colonic epithelium (Koh et al., 2016; Baxter et al., 2019). Thus, the reduction of beneficial SCFA-producing bacteria, as seen in human and rodent HTN (Yang et al., 2015; Gomez-Arango et al., 2016; Kim et al., 2018; Yang et al., 2019b; Calderón-Pérez et al., 2020; Overby and Ferguson, 2021), disrupts the symbiotic host-microbiota relationship leading to disease. In return, the bacteria have most likely evolved by adapting and responding to the host, as evidenced by the effects of genetic host manipulations on gut bacterial composition (Yang et al., 2017; Bartley et al., 2018). Recent studies have attempted to address the complexity of host-microbiota interactions in symbiosis and dysbiosis. Of note, a recent study using isotope tracing found that the host can regulate the composition of gut bacteria by allowing the passage of host-circulating metabolites into the gut (Zeng et al., 2022). These host/gut cometabolites were found to be beta-hydroxybutyrate (BHB), lactate and urea, among other, which are preferentially utilized as nutrients by certain bacterial communities. Future studies should investigate how these cometabolites contribute to regulation of gut microbiota eubiosis and how this interaction reflects on BP regulation. We have recently shown that the circulating BHB and gut microbiota are both salt-responsive (Chakraborty et al., 2018). Moreover, we found that circulating BHB was decreased with high salt feeding, and that supplementation with BHB alleviated salt-sensitive HTN, but the contribution of gut microbiota to BHB generation or the potential direct effect of BHB on gut microbiota in BP regulation remains unknown. Thus, gut microbiota may have coevolved with the host to produce, utilize and respond to a variety of the same metabolite-substrate-effectors, reflected in the expression of some of the same genes by the bacteria and the host (Bartley et al., 2018; Yang et al., 2018, 2022; Hsu et al., 2022). Thus, the utilization of combined omics, employed at both microbiota and host levels, will lead to better predictions and targeting of host-microbiota interactions for therapeutics.

In conclusion, as noted through the numbers of studies reported in PubMed, microbiota is an emerging and important research area in HTN, surpassing that of GWAS and QTL studies of HTN (Figure 1a,b). Although research is still in early conception, given that the gut metagenomes co-evolve with the host and are critical for BP regulation, risk prediction for HTN using a PRS may be more informative in combination with new bacterial analysis approaches leading up to a 'MRS' that encompass both the metagenomic profiles and the functional bacterial readouts. The groundwork required for accumulating metagenomic signatures is admittedly daunting because of the fluctuating nature of microbiota, but collecting these data is important for its ultimate convergence with PRS for enhancing predictive strategies for HTN. Such an endeavor demands intense computational analyses that may only be addressable with AI strategies.

## An application of AI and machine learning in HTN research

AI refers to methods for transferring human intellect to computers that can stimulate human learning and thought processes by using sophisticated algorithms and powerful computing capacity to process large amounts of data (Chaikijurajai et al., 2020; Tsoi et al., 2021). Machine learning (ML) and deep learning (DL) are the two subclasses of AI (Tsoi et al., 2021). ML finds the association between the provided training datasets with variables and then performs predictive analyses on the new sets of data (Tsoi et al., 2021). ML is

further classified into supervised and unsupervised learning (Chaikijurajai et al., 2020). Supervised ML relies on the labeled input–output paired data which is then used for the prediction of known output (Cheng et al., 2011). It employs a variety of methods, including neural networks, support vector machines, random forest and naive Bayes (Cheng et al., 2011). On the other hand, unsupervised ML employs unlabeled datasets to predict unknown outputs by detecting underlying patterns or correlations among the variables (Cheng et al., 2011; Chaikijurajai et al., 2020). The principal use of DL is pattern recognition, such as in voice and image analysis (Chaikijurajai et al., 2020).

AI is increasingly being used in human HTN research (Figure 1b). Recent studies have shown how AI has the ability to reduce the worldwide burden of HTN and promote the development of HTN-related precision medicine (Golino et al., 2014; Ye et al., 2018; Lacson et al., 2019; Kanegae et al., 2020; López-Martínez et al., 2020; Soh et al., 2020; Schrumpf et al., 2021). As a result, the main goal of these investigations is to enhance the clinical management of HTN. Persell et al. conducted a randomized clinical trial of 297 persons with uncontrolled HTN to examine the impact of AI, smartphone coaching apps monitoring systolic BP and HTN-associated behavior. At the 6-month follow-up, the researchers did not discover decreased BP, but they did create a space for the possibility of different treatment effects among age subgroups (Persell et al., 2020). Pan et al. (2019) coupled auscultatory waveforms data with ML to measure BP from Korotkoff sound recordings and examine the impact of movement disturbance on BP regulation. Among 40 healthy volunteers, their brand-new DL-based automatic BP measurement instrument showed encouraging accuracy in BP monitoring both when resting and not resting (Pan et al., 2019). In 965 participants, Li et al. employed ML to identify genetic and environmental risk factors for HTN. To identify risk factors for HTN in the Northern Han Chinese population, they created two separate models for systolic BP (composed of age, body mass index, waist circumference, exercise [times per week], parental history of HTN [either or both], and 1 SNP (rs7305099)) and diastolic BP {composed of weight, drinking, exercise [times per week], trigly-ceride, parental history of HTN [either or both] and 3 SNPs (rs5193, rs7305099, rs3889728)} with an AUC (area under the curve) of 0.673 and 0.817 for the systolic BP and diastolic BP models respectively (Li et al., 2019a). Future use of these AI/ML technologies to HTN may be combined to create a 'clinical risk score' (CRS).

To investigate the multifactorial causes of high BP, Louca et al. recently combined environmental, dietary, genetic, metabolite, biochemical and clinical data from two different cohorts. Then they applied the ML XGBoost algorithm to this multimodal domain. They included 4,863 TwinsUK subjects for the study and used 2,807 subjects from the Qatari Biobank to validate their findings. They discovered 30 overlapping features between the two groups, including age, BMI, sex, dihomo-linolenate, urate, *cis*-4-decenoyl cartinine, lactate, glucose, cortisol, chloride, histidine and creatinine to be associated with HTN. These BP biomarkers are crucial for prioritizing mechanistic investigations and for finding effective novel therapies for HTN (Louca et al., 2022). Although this research examined a number of significant CRS and PRS domains to pinpoint the critical elements involved in the regulation of BP, gut microbiota features, which are crucial for building MRS, were not taken into account. Nakai et al. in their recent study performed the first gut microbiome multisite study involving 70 human subjects with HTN and without HTN. The

authors combined ML with microbiome pathway analysis and reported differential microbial gene pathways between hypertensive and normotensive participants despite similar gut microbiota profile (Nakai et al., 2021).

As one of the initial steps toward application of AI/ML in development of a MRS for HTN, our group had recently interrogated if the composition of microbiota may be used to classify patients with or without CVDs (Aryal et al., 2020). Due to the lack of information on the status of HTN in the American Gut Project, we resorted to classification of a broader group of patients. Using the top operational taxonomic unit features obtained from fecal 16S ribosomal RNA sequencing data of 478 CVD and 473 non-CVD human subjects, random forest, a supervised ML algorithm, was able to correctly classify between patients with CVD and without CVD, with an AUC value of 0.70. It denotes the prospective capacity of ML for case and control distinction (Aryal et al., 2020). Considering the wide range of variability in binning CVD as a single phenotype, an AUC of 0.7 further signifies that microbiota contribute to CVD, and that an association between disease and microbiota can be identified using AI. To move closer to the eventual objective of creating MRS for HTN, such data are required in the context of HTN.

Beyond its usage in healthcare, AI/ML can be used to understand GWAS results by spotting intricate underlying data patterns that make predictions easier. Such methodology improved the prediction of PRS for height, body mass index and diabetes (Paré et al., 2017). Since there exist high-quality GWAS data for HTN, there is a possibility that similar AI/ML methodologies will be merged with CRS and MRS to improve the translational capacities of PRS for HTN (Figure 2).

## Limitations of AI in HTN research

Although the use of AI in HTN has the potential to revolutionize risk prediction, this goal has significant constraints as listed below: (i) There are currently no standards for reporting AI investigations in HTN cases with sufficient rigor. In many publications, for instance, external validation datasets are not used. Very few research articles report the model calibration metrics and, bias brought about by algorithms is typically disregarded (Du Toit et al., 2023). (ii) There is a paucity of open-access databases that provide information on the genotypic and phenotypic characteristics of HTN. (iii) a major current limitation is that large cohort data containing both genomic and microbiota data are lacking. (iv) AI/ML operates in a 'black box' (i.e., it is unclear how it does what it does), which is claimed to be the main reason why physicians are reluctant to implement AI technology in clinical practice (Cheng et al., 2011). (v) the interpretability of the AI models, the absence of cause-and-effect reasoning, the capacity to self-monitor errors, and the presence of societal biases are a few more drawbacks (Padmanabhan et al., 2021).

Some solutions for the limitations mentioned above could be to (i) develop easily interpretable AI models which can discern the relationship between the variables contributing to HTN, (ii) promote initiatives for setting up large-scale and rapid data-

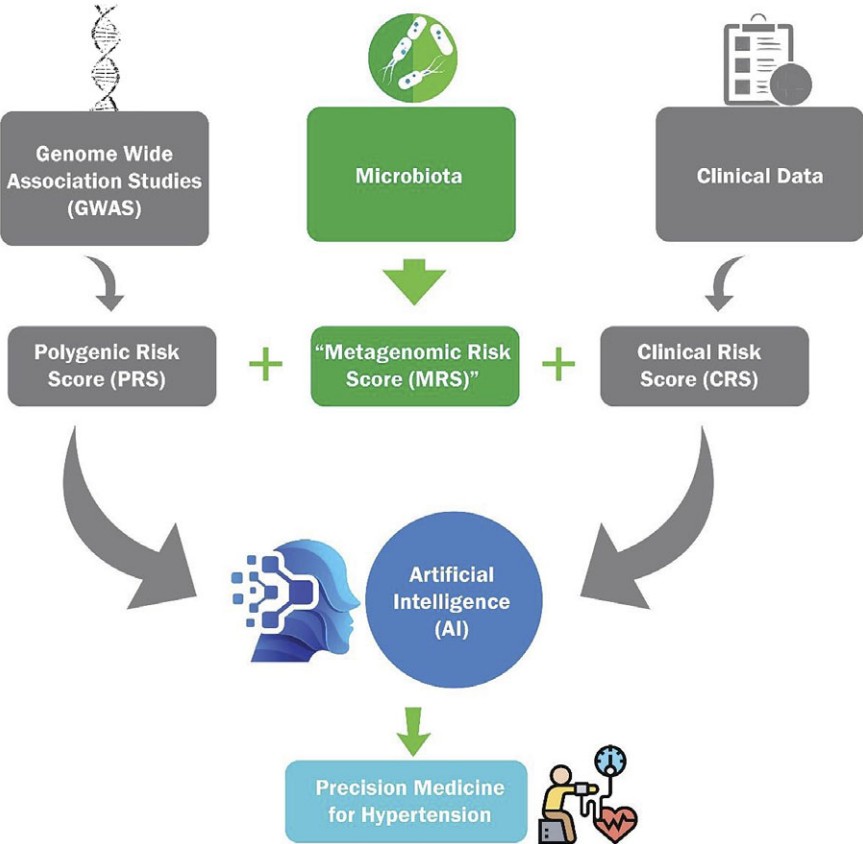

**Figure 2.** The integration of polygenic risk score, metagenomic risk score and clinical risk score using artificial intelligence is required for the precision medicine in hypertension.

sharing of large cohort data specifically pertinent to HTN and (iii) development of standardized methodologies to control for rigor of human judgment into the AI systems for determining errors (Padmanabhan et al., 2021).

In conclusion, this review has summarized the mounting evidence that BP is closely correlated with the microbiota, which make up the second-largest genome after the host genome. In light of this borgeoning evidence, we propose exploiting such data for the development of a MRS as a predictive index for HTN. Additionally, we propose using MRS as part of a larger framework that incorporates PRS and CRS to build an AI-based model. Considerable research efforts to generate MRS may serve as a tool to enhance the existing, primarily insufficient predictive capability for the management of HTN.

**Open peer review.** To view the open peer review materials for this article, please visit http://doi.org/10.1017/pcm.2023.13.

**Data availability statement.** Data availability is not applicable to this article as no new data were created or analyzed in this study.

**Acknowledgments.** The authors thank Mr. Sarbesh Rijal, MS, Nepal for generating Figure 2. Graphical abstract was created with BioRender.com.

**Author contribution.** Conceptualization: S.A., I.M., B.J.; Drafting the original manuscript: S.A., I.M., B.S.Y., J.Z., D.J.D., M.V.-K., B.J.; Revision, editing and final approval of the manuscript: S.A., I.M., X.M., B.S.Y., R.T., P.S., I.O., J.Z., D.J.D., M.V-K., B.J.

**Financial support.** This work was supported by the National, Heart, Lung, and Blood Institute (NHLBI) (B.J., R01HL143082); the National Institutes of Health (NIH)-National Cancer Institute (NCI) grant (M.V.-K, R01CA219144); NHLBI and Research Corporation for Science Advancement (D.J.D., R01HL134838, RCSA27917); NHLBI (J.Z., R01HL152162); NHLBI (I.O., R00HL153896); American Heart Association Career Development Award (P.S., 855256) and American Heart Association Award (B.S.Y., 831112).

**Competing interest.** The authors declare none.

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
