## [Reviewer Report]

*Comments to Author*: In this review, Sachine and collaborator first report on studies using the polygenic risk score to predict hypertension, they then discuss some of the recent literature on the role of the gut microbiome on HTN regulation and they conclude highlighting the values of AI studies. Though the paper is a broad literature review including many topical papers, the overall focus and main take home message is not clear and the flow is not always there. In particular the section on AI is not very well incorporated. 

Impact statement and abstract are poorly written. There are several typos and grammatical mistakes (this is a problem throughout). Also, some sentences are totally unclear (eg lines33-37). Lines 67-71 in the abstract do not mirror what is then written in the introduction.

The gut microbiome part is heavily biased on animal studies. This is to be expected as so far, not many human studies have been conducted. However, I would shorten the animal part and perhaps also discuss the few population based human studies comparing HTN cases and normotensive controls besides the dietary intervention studies.

The section on F Prau as a novel probiotic for CKD is well written, but its relationship with HTN, from what is reported, is far fetched. Can the authors elaborate and discuss papers where the association between F Prau and HTN/BP is reported (eg PMID: 28884091).

I agree with the authors on the importance to investigate microbial metabolites . Could the authors expand on what microbial metabolites have been identified to associate with HNT /BP?

In the AI section, the authors should also discuss some of the more recent literature:

eg doi: 10.1161/hypertensionaha.121.17288 

https://doi.org/10.1016/j.ebiom.2022.104243

Finally, I believe one of the main limitation of AI/ML is the lack of (many) large cohorts that currently have genome and microbiome data available

---

## [Editor Report]

*Comments to Author*: The review will be strengthened with a bit more human focus and addressing the comments from the reviewer.